# Safety and Effectiveness of a New Electrical Detachable Microcoil for Embolization of Hemorrhoidal Disease, November 2020–December 2021: Results of a Prospective Study

**DOI:** 10.3390/jcm11113049

**Published:** 2022-05-28

**Authors:** Miguel Angel De Gregorio, Román Bernal, Juan Jose Ciampi-Dopazo, José Urbano, Alfonso Millera, José Andres Guirola

**Affiliations:** 1Interventional Radiology, Hospital Clínica Quirón Salud, 50006 Zaragoza, Spain; 2Grupo de Investigación en Técnicas Minimamente Invasivas (GITMI), University of Zaragoza, 500013 Zaragoza, Spain; troech1286@gmail.com; 3Interventional Radiology, Hospital Virgen de las Nieves, 18001 Granada, Spain; juanciampi@hotmail.com; 4Interventional Radiology, Hospital Universitario Ramon y Cajal, 28001 Madrid, Spain; jurbano34@gmail.com; 5Department of Surgery, Hospital Clínico Universitario Lozano Blesa, 50009 Zaragoza, Spain; millera@unizar.es; 6Interventional Radiology, Hospital Clínico Universitario Lozano Blesa, 50009 Zaragoza, Spain; joseandresguirola@gmail.com

**Keywords:** catheter-directed hemorrhoidal dearterialization (CDHD), internal hemorrhoids, treatment

## Abstract

Purpose: The purpose of this study was to prospectively evaluate the efficacy and safety of a new, bare platinum, detachable microcoil as a metallic embolization agent in the treatment of hemorrhoidal disease. Material and Methods: This prospective single-center study evaluated a new, bare platinum, electrical, detachable microcoil (Prestige plus coil (Balt Montmorency France)) for use in vascular embolization in patients with hemorrhoidal disease. Between January 2020 and January 2021, 24 embolization procedures were performed in 21 patients (12 males, 9 females; mean age 44.3 ± 7.3). The inclusion criteria were: (a) participants with grade I, II and III hemorrhoidal disease on the Goligher classification; (b) patients older than 18 years of age with a score of greater than 4 on the French bleeding score (FBS) scale; (c) patients with scores greater than 2 on the scale of discomfort proposed by Tradi and Farfallah. (d) patients who underwent treatment that included the use of the new novel coil (Prestige plus coil (Balt)) as an embolic material. The exclusion criteria were participants who failed to provide informed consent and participants diagnosed with rectal bleeding due to other causes (cancer, fissures or others). Participants with severe renal insufficiency, non-correctable coagulation abnormalities and adverse reactions to the contrast medium not correctable with medication were also excluded. The symptoms, technical aspects, the transarterial approach, clinical and technical success complications and short-term outcomes were assessed. Results: Technical success was obtained in 100% of the cases. Seventeen (80.9%) patients experienced improvements in their hemorrhoidal disease. The VAS and QL scores improved by 4 and 1.5 points (81.2% and 87.5%), respectively, after embolization (pV: 0001). Three (14.2%) patients underwent a second embolization due to rebleeding. One patient (4.7%) underwent surgery. No major complications were observed. Three patients had minor complications. The assessment of subjective post-treatment symptoms and QL surveys showed significant differences from the baseline survey. Likewise, the measurement of the degree of satisfaction using a telephone survey at 12 months revealed a high degree of patient satisfaction over 10 points (mean 8.3 ± 1.1). Conclusions: The present study demonstrates that the use of the new, platinum, detachable, electrical microcoil is safe and well-tolerated in the treatment of hemorrhoidal disease. Key points: Catheter-directed hemorrhoidal dearterialization (CDHD) is the procedure of embolization with embolic agents for the treatment of internal hemorrhoids. CDHD is a simple and safe procedure that is accepted by patients and preserves the anal sphincter; it presents few complications when metal devices or microspheres are used as embolic agents. As the recommended embolization agent in treatments, the Prestige electrical, detachable coil is a safe, easy-to-use and effective arterial embolic device.

## 1. Introduction

Hemorrhoidal diseases are the most common anorectal conditions, affecting 4–35% of the population, and are a major cause of clinically significant anorectal bleeding. Patients who have these diseases also experience substantial negative impacts on their quality of life [1]. The coloproctologist, Varut L., defined hemorrhoidal disease as the symptomatic enlargement and/or distal displacement of the anal cushions, and in addition to an abnormally dilated vascular channel and destructive changes in the supporting tissue within the anal cushions, there is emerging evidence that suggests hemorrhoidal disease is associated with a hyperperfusion state in the anorectal region and some degree of tissue inflammation [2]. WH Thomson [3], basing himself on cadaver studies, described for the first time, the vascularization and the pathophysiology of hemorrhoids, he observed that terminal branches of the superior rectal artery (SRA) contribute to the vascularization of the corpus cavernosum rectum. In 2006, Aigner F et al. [4] demonstrated that both submucosal and transmural branches play an essential role in the blood supply to this rectal area. According to Golingher [5], hemorrhoids can be categorized into four grades. Grade I hemorrhoids are nearly normal in appearance and do not prolapse. Grades II, III and IV exhibit bleeding and prolapse with different types of reducibility: grade II hemorrhoids are spontaneously reducible; grade III hemorrhoids are digitally reducible; grade IV hemorrhoids are not reducible.

Bleeding in various amounts is the main symptom of hemorrhoidal disease (HD). Other common signs and symptoms include burning, pain, itching, discomfort, a feeling of pressure and the development of skin tags [6]. The therapeutic options vary according to the evolutionary degree of the hemorrhoid. Conservative procedures are generally reserved for lower-grade hemorrhoids (I–II), while higher grades (III–IV) require surgery [7]. Hemorrhoidectomy is considered the standard treatment for hemorrhoids by Milligan Morgan [8]. At present, other surgical treatments have emerged, such as Doppler-guided hemorrhoidal ligation (DgHAL) [9], as well as the Doppler technique of transanal hemorrhoidal dearterialization (THD) [10].

In 2014, based on dearterialization, Vidal et al. described a technique used to occlude hemorrhoidal arterial blood flow called the smear technique [11]. This technique was used in three patients diagnosed with bleeding HD and achieved satisfactory results. In addition, in 1994, Galkin EV [12] published an article that concerned embolization in 34 patients that did not experience recurrences. 

Coils and microparticles or spheres are the most widely used embolizing agents, although, in theory, other agents could be used. Nevertheless, Tradi F et al. [13] recommend that embolic fluids should be studied closely before being used, especially as microcoils are mostly very effective and safe. There are 47 types of coils on the market made by nine companies [14]. Most are platinum; 20 (42.5%) are removable, and 27 are pushable. The new microcoil, the Prestige Plus coil (Balt), provides good deliverability and compliance combined with its ease of delivery and reliable detachment.

We present our results from this non-randomized, prospective, single-center study in 26 patients with grade I, II and III hemorrhoidal disease.

## 2. Materials and Methods

### 2.1. Patient Inclusion

This prospective non-randomized, single-center study was approved by the Ethics Committee of our CEICA region, CP.CI. PI21/008.

All the patients included in the study met the inclusion criteria and provided informed consent. According to the Guidelines on the management and treatment of hemorrhoidal disease of the Italian Society [15]. “Embolization was used in patients with disabling and refractory hemorrhoidal symptoms and without irreducible prolapse (Level of evidence: 2; Grade of recommendation: C”. Treatment indications were based on the work of Vidal et al. [11] and Galkin EV [12].

The following inclusion criteria were used: (a) patients with grades I, II and III HD, determined using the Goligher classification [5]; (b) patients older than 18 years of age with a score greater than 4 on the French bleeding score (FBS) [16]; (c) patients with discomfort that scored greater than 2 on the discomfort scale proposed by Tradi F. et al. and Farfallah N. et al. [17,18]; (d) patients who underwent treatment that included the use of the new novel coil (Prestige plus coil (Balt)) as the embolic material. The exclusion criteria were patients who refused to provide informed consent and patients with rectal bleeding due to other causes (cancer, fissures or others).

Patients with severe renal insufficiency, uncorrectable coagulation disorders and adverse reactions to the contrast medium uncorrectable with medication were also excluded.

From January 2020 to January 2021, a total of 21 patients (mean age, 44 ± 7 years (range: 31–66 years); 12 men and 9 women) were recruited to undergo alternative treatment. The mean body mass index (BMI) of the participants was 27 ± 1.3 (range: 25–37). The distribution of hemorrhoids according to the Goligher classification is as follows: grade I, 3 patients (14.2%); grade II, 15 patients (71.4%); and grade III, 3 patients (14.2%) (Table 1).

The severity of bleeding was measured according to the FBS [16,17,18,19]. The FBS is scaled from 0 to 9, based on three separate variables: the frequency of bleeding (0–4), the type of bleeding (0–3) and the presence of anemia (0–2). The patients were evaluated and referred by a coloproctologist. However, all the patients were evaluated in an interventional radiology outpatient clinic to explain the procedure to them and allow them to provide informed consent and review the CIRSE checklist [20].

### 2.2. Outcome Measures and Definitions

The primary objective of this study was to prospectively assess the safety and feasibility of treating grade I, II and III hemorrhoids with superior rectal artery (SRA) embolization in a cohort of twenty-one patients. Technical success was defined as the occlusion of all the SRA in its distal segment. To define clinical success, three different evaluations were performed at follow-up after the first month: rectal bleeding measured with the FBS, pain measured with the visual analogue score (VAS) scale and quality of life (QOL) measured using the scale proposed by Tradi et al. and Farfallah N. et al. [17,18].

To assess the clinical outcomes, five procedural outcomes were defined: 1, healing (FBS = 0, VAS = 0, QOL = 0); 2, improvement (B 1–3, V < 3, D < 2); 3, worsening (B > 3, VAS > 3, QOL > 3); 4, no change (values similar to baseline); and 5, the recurrence of bleeding. Bleeding recurrence was classified into three grades: mild (bleeding frequency 3–4), moderate (bleeding frequency 1–2) and unchanged compared to the time before treatment. Complications were reported according to the classification of the Society of Interventional Radiology [21] as minor (A–B) and major (C–F) complications.

After one year, all the patients completed unvalidated quality of life and satisfaction surveys. Both were subjectively scored from 0 to 10, where a score of 0 represented patients who subjectively reported being in poor health in general and very dissatisfied with the procedure, and a value of 10 represented patients who were in very good health and very satisfied. Possible vascular access complications (hematoma and ischemia of the hand) were evaluated according to the Common Terminological Criteria for Adverse Events, version 5.0 [22].

### 2.3. Coil Characteristics

In all cases, Prestige plus coils (Balt, Montmorency, France) were used. These platinum coils are electrically detachable and have adequate organoleptic and navigation characteristics, which allow them to be handled easily and mean they are a reliable and fast form of treatment.

### 2.4. Embolization Technique

All the procedures were performed on an outpatient basis with a stay in the preparation room next to the operating room for 2–4 h. Antibiotic or analgesic premedication was not routinely provided.

In all cases, vascular access was obtained through the left radial artery (100%; 21/21). In all cases, the Barbeau test was performed. Radial access was not achieved in cases where the wave did not return to a normal amplitude or was abnormal after 2 min of radial compression. Radial access was also not achieved in patients who had a radial artery diameter of less than 1.8 mm. In the event that they were not candidates for the radial approach, these patients were withdrawn from the study.

To access the radial artery, 9 mL of 1% lidocaine with 100 μg of nitroglycerin was administered into the subcutaneous tissue near the artery. An access set for micropuncture (Cook Medical, Bloomington, IN, USA) was used, consisting of a 21 G × 7 cm echogenic needle and a 0.018” × 40 cm guidewire. 

The radial introducer was 5 F in diameter and 10 cm long (Radiofocus, Terumo, Tokyo, Japan). To prevent arterial spasm or thrombosis, a total of 2.5 mg of verapamil, 200 μg of nitroglycerin and 3000 IU of unfractionated heparin mixed with 20 mL of the patient’s own blood obtained using the radial approach, was administered immediately after the artery was accessed using the introducer sheath.

Using a 0.035-inch, 150 cm hydrophilic guidewire (Radiofocus, Terumo, Tokyo, Japan) and a 5F, 125 cm MPA catheter (Cook Medical, Indiana, USA), the abdominal aorta was advanced, and the proximal segments of the inferior mesenteric artery and the superior rectal artery (SRA) were selected. A selective angiography of the SRA was performed by administering 12 mL and 4 mL/s of iodinated contrast (Optiray 320 Guerbet BP 57400) to determine the anatomy of the SRA and possible communication with the middle or inferior rectal artery. A 150 cm Progreat 2.4 or 2.7 Fr microcatheter (Radiofocus, Terumo, Tokyo, Japan) was placed as close as possible to the corpus cavernosum recti (Figure 1).

In all the cases, Prestige coils (Balt) with diameters of 2–5 mm and various lengths (10–40 cm) were used. Prestige microcoils were also used in re-embolization procedures in rebleeding patients and, if necessary, 500–700 µm Tris-acryl Gelatin microspheres (TAGM) (Merit Medical Spain) were used.

After the procedure, the microcatheter, the MPA catheter and the introducer were all carefully removed. The access point was manually compressed for at least 10 min, or a radial compression device (PreludeSYNC™ Radial Compression Device—Merit Medical, Utah, USA) was used. Data concerning the total radiological examination time, air kerma, fluoroscopy time and product area dose were collected (Figure 2).

### 2.5. Care during and after the Procedure

During the procedure, the patients were monitored via electrocardiography, blood pressure, heart rate and pulse oximetry. When the patient experienced anal discomfort during SRA embolization, 20 mg of scopolamine butylbromide mixed with 100 mL of saline was slowly intravenously administered.

After the procedure, the patient returned to the preparation room, where the patient remained for at least two hours under monitoring. Patients were discharged after they exhibited normal vital signs (respiratory and cardiac), walked, and were not in substantial pain. Non-steroidal anti-inflammatory drugs were prescribed for two or three days (scopolamine butylbromide and metamizol). A diet rich in fiber, vegetables and plenty of fluids was recommended for a few days to prevent constipation.

The patients were followed up for one month at the interventional radiology and coloproctology outpatient clinic. Follow-up was then performed at 3, 6 and 12 months, involving an anoscopy and the examination of clinical findings.

Before and after (between 3 and 6 months), the embolized patients voluntarily answered a questionnaire regarding their symptoms and quality of life/discomfort. The questionnaire created by Rørvik HD et al. [23] was translated into Spanish and modified to assess symptoms in Spanish-speaking patients regarding health-related and hemorrhoidal-disease-related quality of life. At the end of the study, a satisfaction telephone survey was carried out, which was scored from 0 to 10. Scores from 0 to 3 indicated that patients were not at all satisfied; scores from 4 to 6 indicated they were somewhat satisfied; scores from 6 to 8 indicated they were quite satisfied; scores from 9 to 10 indicated they were very satisfied.

### 2.6. Statistical Analysis

The qualitative variables are expressed as frequencies and percentages. Comparisons between groups were made using contingency tables with Pearson’s χ^2^ or Fisher’s test, depending on the magnitude of the expected frequencies. The quantitative variables are expressed as mean ± standard deviation.

## 3. Results

Follow-up was performed with the clinical assessment of symptoms and anoscopy at 3, 6 and 12 months in 21 patients (100%); 1 patient (4.7%) was diagnosed with anemia. A total of 9 patients experienced tenesmus (42.8 %), perianal pain was recorded in 10 patients (47.6%), and rectal pruritus was recorded in 14 patients (66.6%).

The mean pre-embolization hemoglobin level was in the range of 11.0 ± 1.9 g/dL (9.2–13.4 g/dL). A total of 52.5% of the patients had functional constipation. Two patients (9.5%) had a surgical history for the treatment of anal fissures, and six (37.5%) patients reported comorbidities or associated diseases: three (14.2%) reported diabetes, one (4.7%) reported fibromyalgia, two (9.5%) reported pelvic congestion syndrome and two (9.5%) reported other comorbidities or associated diseases. Two patients (9.5%) were being treated with oral anticoagulants.

All the patients underwent a diagnostic anoscopy that demonstrated dilated and congestive internal hemorrhoids in all cases.

All the patients were classified using the Goligher classification from grade I (3 patients (14.2%)), grade II (15 patients (71.4%)), grade III (3 patients (14.2%)) and grade IV (no patients).

A total of 21/21 (100%) patients underwent SRA embolization during the first embolization. A total of 3/21 (14.2%) required a second embolization due to rebleeding. In two patients (9.5%), the MRA was embolized, one with coils and TAGM of 500–700 µm, and the other was embolized with coils alone. The result was satisfactory in all the patients. One patient (4.7%) underwent surgery.

No major complications were observed. Three patients had minor complications (one case of radial hematoma and two cases of minor tenesmus).

On average, 7.8 ± 1.5 electrical detachment coils were used per patient (range 6–12) (Table 2 and Table 3).

The mean procedure time was 44.3 ± 11 min, the mean fluoroscopy time was 37.5 ± 11.1 min, the average product radiation dose was 242,364.3 ± 160,322 mGycm^2^ and the average air kerma was 231.2 ± 213 mGycm^2^ (range: 207–237 mGcm^2^).

### Follow-Up

When a patient did not attend the monthly outpatient consultation, they were called by phone to determine their clinical status and other outcomes. At 12 months, the clinical outcomes were: healing (18/21 (85.7%)), improvement (2/21 (9.5%)) and no changes (1/21 (4.7%)) (Table 4).

Figure 3 shows the results from 0 to 10 points of the modified Rorvik survey in which they ask about symptoms and health data, about quality of life. Additionally, in the same figure, you can see data relative to the degree of satisfaction of the telephone survey carried out once the study was completed at 12 months.

## 4. Discussion

The purpose of this study was to evaluate the use of a new coil in emborrhoid procedures and its clinical results. Since the first publication [11] that concerned the emborrhoid technique by Vidal et al., his group of researchers and other authors have published many studies evaluating this procedure in patients with hemorrhoidal diseases. However, most published series were carried out on few patients (with most having less than 50 patients) and with a short follow-up time of 12 months or less.

In most of the published series, technical success rates of 90–100% have been reported, while clinical success rates range from 63% to 97% [11,12,16,17,18,19,23,24,25,26,27,28,29,30,31,32,33,34,35]. However, it is difficult to compare the results from the different series as there is no single and homogeneous criterion and standard that defines clinical success [7]. In the present cohort, after one year, for the 21 treated patients, the improvement rate was 87.5% (cure). Two patients (9.5%) presented improvements in symptoms with isolated mild symptoms, and one patient showed no change with respect to the previous initial symptoms they exhibited.

According to Milligan Morgan and Ferguson, hemorrhoidectomy is the gold-standard treatment that achieves higher clinical success rates than embolization, but it also can cause complications and requires hospitalization [36]. In our series, only one patient in whom embolization had failed required conventional surgery; no other alternative techniques, such as Doppler-guided hemorrhoidal dearterialization (THD or DGHAL), were performed [37,38]. Generally, the emborrhoid technique is recommended in patients in which surgery is not recommended. However, in our work, the majority of patients chose the interventional procedure as an alternative to surgery. This was facilitated by the support of the Coloproctology department.

This study had a follow-up period of 44 ± 29 months. No major complications were observed. Three patients (14.2%) had minor complications (one case of radial hematoma and two cases of anal tenesmus that did not require admission or additional treatment). The mean hospitalization time was 1 ± 0.2 days. Rectal bleeding was observed as recurrent in seven (33.3%) patients, but in four patients, it was mild and isolated Only three patients (14.2%) required a second embolization.

Our study presents improvements upon the results obtained with DGHAL in which a recurrence of up to 30% at 1 year [37,39] was shown and which were similar to those obtained with THD by Ratto et al. [10]. There are no major differences when comparing the use of THD and catheter-directed hemorrhoidal dearterialization (CDHD) [7] with regard to clinical success and recurrence. Regarding the use of DGHAL and THD, the clinical success rate was 82.3–95.7% and the recurrence rate was 9.5–30%, while when embolization (CDHD) was used, the clinical success rate was 78.9% (66–96%) and the disease recurrence rate was 22% (5–43%).

In general, the use of DGHAL and CDHD did not modify or improve external hemorrhoids. In the case of DGHAL, after transanal dearterialization treatment, a mucopexy can be performed on external hemorrhoids. The use of embolization does not improve prolapse, but some authors [16,19] believe that prolapse may improve over time due to decongestion by decreasing arterial flow after embolization. This phenomenon has also been observed in cases of the use of DGHAL without performing mucopexy.

By contrast, embolization treatment offers advantages over others, such as rubber band ligation (RBL) and injection sclerotherapy (IS), which can present post-procedure complications between 1–50% of RBL cases. Pain has been observed in 8–80%, and severe pain in 4–20% [40]. In post-embolization treatment, there is usually no post-procedure pain.

In our series, prolapse before and after embolization was not evaluated. The patients were already aware that the prolapse required other treatment and it was not included as an indication of embolization. Data in favor of CDHD suggest that it is a treatment that can be performed on an outpatient basis, is slightly or not at all painful and has few complications.

Radial access is one of the factors that allow outpatient treatment, in addition to the fact that it causes less dissatisfaction for the patient and creates fewer complications [41]. In our series, 100% of patients were treated with radial access, with 4.7% experiencing minor complications and no increase in procedure time or radiation dose being observed.

In our series, the technical success rate was 100%, with a 14.2% rate of minor complications according to the classification of the Society of Interventional Radiology [21]. There were no major complications, finding consistent with that reported in recent publications in the medical literature [11,16,17,18,24,42,43,44]. However, recently, Eberspacher Ch [45] reported a case study of rectal sigmoid ischemia secondary to hemorrhoidal microparticle embolization in a 58-year-old patient.

In this sense, Tradi F et al. [13] advise that other embolizing agents should be studied closely before being used, such as embolic fluids, especially as microcoils, which have been shown to be very effective and safe. In our experience, the microcoil Prestige Plus (Balt, Montmorency, France) is a new, novel coil with good deliverability combined with reliable detachment. We did not experience any technical failures during our series of procedures. Additionally, a report [13] indicated that ethylene vinyl alcohol copolymer may be an unsafe agent for hemorrhoid embolization, as in their experimental study in pigs, necrosis of the distal rectum with mural infarction was observed in animals that underwent embolization with a copolymer of ethylene vinyl alcohol.

Other authors reported pain and tenesmus rates between 4.6 and 15% and 54% with the use of polyvinyl alcohol (PVA) and TAGM as embolizing agents [19,24,25,26,27,28].

Iezzi R et al. [41] recommend the radial approach, as it is a safe mode of access that allows the outpatient management of patients.

This study has some limitations: it was a retrospective study with a prospective database, it was a non-randomized study, the study population was small and the follow-up period was only one year.

In conclusion, CDHD is a feasible procedure; it does not alter the functionality of the sphincter; it is relatively simple, safe and painless, and it does not require hospitalization. The electric detachable prestige coil is a safe and effective device; it is easy to handle, and it exhibits great control during its release. In our study, a significant percentage of patients experienced hemorrhoid resolution after the embolization procedures. Future evaluations that compare embolization techniques with other procedures for the treatment of hemorrhoids are warranted [46,47].

## Figures and Tables

**Figure 1 jcm-11-03049-f001:**
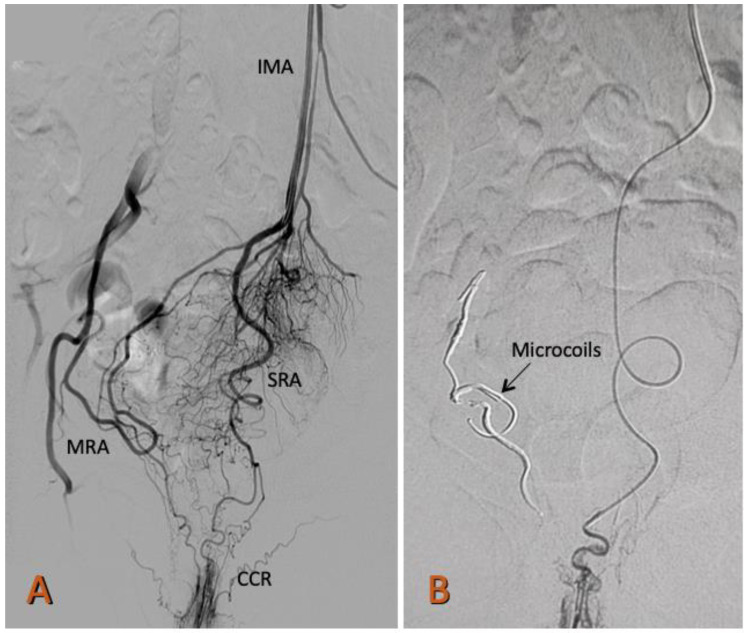
Arteriography of the superior rectal artery (SRA) from the inferior mesenteric artery (IMA). (**A**) The highly developed right branch of the SRA that irrigates the corpus cavernosum rectum CCR is observed. From the origin of the SRA, a collateral arises that anastomoses with the middle rectal artery (MRA), a branch of the right internal iliac artery. The left branch of the SRA is underdeveloped. (**B**) Occlusion with Prestige coils of the right middle rectal artery (MRA) and the right superior rectal artery (SRA).

**Figure 2 jcm-11-03049-f002:**
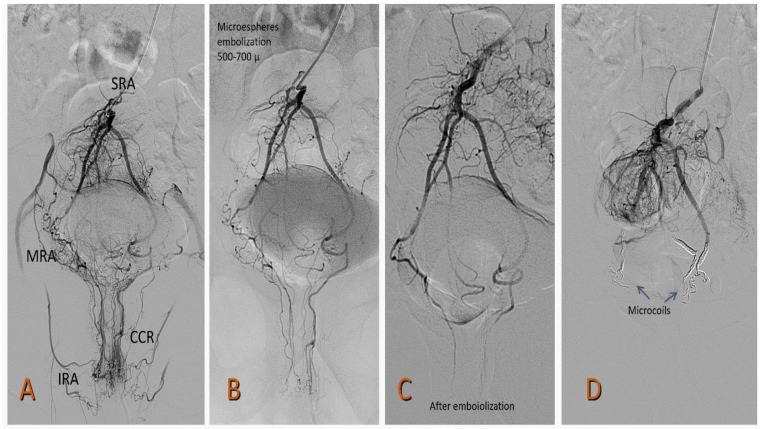
A 49-year-old patient with grade II HD (Goligher) and French bleeding score (FBS) of 6: (**A**) Selective arteriography from the inferior mesenteric artery in which asymmetry of both superior rectal arteries is observed. The right SRA is highly developed with communication with the MRA and IRA. Both provide a lot of risk to the CCR. The left SRA is poorly developed. (**B**) Embolization with 500–700 µ microspheres (Merit Medical). (**C**). Acceptable result after embolization. (**D**) Distal embolization with Prestige microspheres (Balt medical). SRA: superior rectal artery, MRA: middle rectal artery, IRA: inferior rectal artery, CCR: corpus cavernosum rectum.

**Figure 3 jcm-11-03049-f003:**
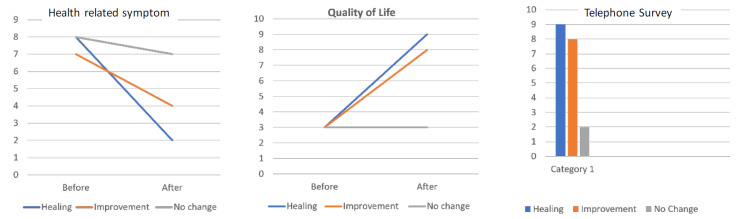
Results of the survey of health and symptoms, quality of life and degree of satisfaction by the telephone survey.

**Table 1 jcm-11-03049-t001:** Months in relation to bleeding (FBS), pain measured using VAS and quality of life (QOL).

Patient Characteristics	
Characteristics	
Age	44 ± 7.3 (range 31–66)
Sex Male Female	12(57.1%) 9(42.8 %)
BMI	27 ± 1.3 (Range 25–37)
Main symptoms French bleeding score Anal Pain Quality Life	5.3 ± 1.1 (range 0–9) 4.7 ± 1.1 (range 3–7) 2.0 ± 0.7 (range 1–4)
Anticoagulants	2 (9.5%)
Grade of prolapse (Golinger) Grade I Grade II Grade II Grade IV	3 (14.2%) 15 (71.4%) 3 (14.2%) 0
Comorbilities Diabetes Fibromyalgia Pelvic congestión symdrome Others (atrial fibrillation, DVT) None	3 (14.2%) 1 (6.5%) 2 (12.5%) 3 (14.5%) 10 (47.6)

**Table 2 jcm-11-03049-t002:** Immediate results after embolization.

	N	Percentage	Cumulative Percentage
**Improvement after embolization**			
No rectal bleeding	13/21	61.90	68.7
Rectal bleeding with improvement	4/21	19.04	80.9
Total	17/21	80.95	
**No improvement after embolization**			
Rectal bleeding and second embolization	3/21	14.28	
Rectal bleeding and surgical treatment	1/21	4.76	
Total	4/21	19.04	

**Table 3 jcm-11-03049-t003:** Results after 12 months in relation to bleeding (FBS), pain measured using VAS and quality of life (QOL).

Follow-Up Scores	Before Treatment	After Treatment	Change (95% CI)	*p* Value
French Bleeding (FB)	4.9 ± 1.3 range (0–9)	0.7 ± 1.5 range (0–6)	4.2 (5.4–1.1)	0.0001
VAS	4.9 ± 1.1 range (3–7)	0.7 ± 1.1 range (0–6)	4.2 (4.9–1.3)	0.0001
Quality life	2.2 ± 0.7 range (1–4)	0.5 ± 0.7 range (0–3)	1.7 (2.2–0.7)	0.0001

**Table 4 jcm-11-03049-t004:** Outcome results after 12 months.

Follow-Up 12 Months	#	%
Healing	18/21	85.7
Improvement	2/21	9.5
No change	1/21	4.7

## Data Availability

This statement if the study did not report any data.

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
