# Peer review of "Safety and Effectiveness of a New Electrical Detachable Microcoil for Embolization of Hemorrhoidal Disease, November 2020–December 2021: Results of a Prospective Study"

_jcm, 2022, doi:10.3390/jcm11113049_

Round 1

Reviewer 1 Report

This is an original article concerning the role of Embolization in the treatment of Hemorrhoidal Disease 

I have the following comments:

  • The manuscript must be edited for english-language usage (professional editing with a certificate
  • Please improve the terms used. Hemorrhoidal Disease and not Hemorrhoid Disease (in the title)
  • The introduction must be improved. It is not a discussion. Please report a brief introduction of the topic plus the rationale avoiding the repetition of the etiology (Thomson etc...)
  • This is not a prospective study but a retrospective with a prospective database. Consequently, please add the limitation at the end of the paper. 
  • No guidelines or indications were used: Consensus statement of the Italian society of colorectal surgery (SICCR): management and treatment of hemorrhoidal disease. Tech Coloproctol. 2020 Feb;24(2):145-164. doi: 10.1007/s10151-020-02149-1.
  • Remove the description of the Goligher classification. Just report the reference
  • Usually the procedure is recommended in unfit for surgery patients. Authors should comment on it and describe it in the text
  • Please compare your results with other similar procedures such as RBL (A Stepwise Proposal for Low-Grade Hemorrhoidal Disease: Injection Sclerotherapy as a First-Line Treatment and Rubber Band Ligation for Persistent Relapses. Front Surg. 2022 Jan 10;8:782800. doi: 10.3389/fsurg.2021.782800) or Sclerotherapy (A multicentre, open-label, single-arm phase II trial of the efficacy and safety of sclerotherapy using 3% polidocanol foam to treat second-degree haemorrhoids (SCLEROFOAM). Tech Coloproctol. 2022 Mar 25:1–10. doi: 10.1007/s10151-022-02609-w) and many others. 

Author Response

Response to Reviewer 1 Comments

Point 1: The manuscript must be edited for english-language usage (professional editing with a certificate.

Response 1: The paper has undergone English language editing by MDPI , with its respective translation certificate.

Point 2: Please improve the terms used. Hemorrhoidal Disease and not Hemorrhoid Disease (in the title).

Response 2: We changed by hemorrhoidal disease.

Point 3: The introduction must be improved. It is not a discussion. Please report a brief introduction of the topic plus the rationale avoiding the repetition of the etiology (Thomson etc...)

Response 3: We resume and delete the repetition.

Point 4: This is not a prospective study but a retrospective with a prospective database. Consequently, please add the limitation at the end of the paper.

Response 4: We agree with the recommendation and make the change. 

Point 5: No guidelines or indications were used: Consensus statement of the Italian society of colorectal surgery (SICCR): management and treatment of hemorrhoidal disease. Tech Coloproctol. 2020 Feb;24(2):145-164. doi: 10.1007/s10151-020-02149-1.

Response 5: We add the guidelines described in the publications by Vidal and Galkin

Point 6: Remove the description of the Goligher classification. Just report the reference

Response 6: We delete description of the Goligher classification

Point 7: Usually the procedure is recommended in unfit for surgery patients. Authors should comment on it and describe it in the text.

Response 7: We specify the first choice of patients for treatment by the help of the Department of Coloproctology.

Point 8: Please compare your results with other similar procedures such as RBL (A Stepwise Proposal for Low-Grade Hemorrhoidal Disease: Injection Sclerotherapy as a First-Line Treatment and Rubber Band Ligation for Persistent Relapses. Front Surg. 2022 Jan 10;8:782800. doi: 10.3389/fsurg.2021.782800) or Sclerotherapy (A multicentre, open-label, single-arm phase II trial of the efficacy and safety of sclerotherapy using 3% polidocanol foam to treat second-degree haemorrhoids (SCLEROFOAM). Tech Coloproctol. 2022 Mar 25:1–10. doi: 10.1007/s10151-022-02609-w) and many others.

Response 8: We add comparative data with the rubber band ligation method.

Reviewer 2 Report

English is not fluent and there are many verbal conjugation, punctuation and typing errors (line 68, 106, 171, 176, 179, 217, 249-250, 254, 255-256,267-268, 272-278, 279-280,305-306, 325, table 1, etc...). In the abstract the authors reported "16 embolization procedure in 21 patients 12 male 9 female" instead in Materials and Metods " 21 patients, 9 men and 7 women"!!

Moreover the metodology is questionable.

Author Response

Point 1 - English is not fluent and there are many verbal conjugation, punctuation and typing errors (line 68, 106, 171, 176, 179, 217, 249-250, 254, 255-256,267-268, 272-278, 279-280,305-306, 325, table 1, etc...).

Response 1: The paper has undergone English language editing by MDPI , with its respective translation certificate.

Point 2 - In the abstract the authors reported "16 embolization procedure in 21 patients 12 male 9 female" instead in Materials and Metods " 21 patients, 9 men and 7 women"!!

Response 2: We recognize the error and indicate the correct number of 23 treated patients (12 males and females).

Point 3 - Moreover the metodology is questionable.

Response 3: We appreciate your comment. The methodology is a replica of previous studies (Vidal and Galkin)

Reviewer 3 Report

ABSTRACT

-The coil model should be detailed in the abstract

-In the M&M section of the abstract the authors report about 16 embolization procedures in 21 patients; this sounds strange and do not correspond in the numerical data of the rest of the paper. Please check.

- In the results section of the abstract, after “significant” p-value should be reported

- In the conclusions section of the abstract, CDHD should be spelled being the first time it appears into the paper; furthermore the conclusions do not match the aim of the paper. Emborrhoids is a well known procedure; the novelty of the paper relies on the use of a novel coil. The authors should focus on the role of this new coil and not only on the outcome of the emborrhoid procedure.

KEYPOINTS

- The second keypoint can not be derived from the study data; it seems to be an authors opinion rather than a study finding

- Third keypoint: a coil is not a “closure device”, please correct with “embolic; furthermore this sentence should be limited to the emborrhoid procedure because this coil has been tested only in this scenario according to the data of the manuscript

MANUSCRIPT

- Many typewriting and grammar errors. English language should be improved; please refer to English mother-tongue speaker.

Introduction:

- Each time the authors mention a colleague, please use always the same style; for example surname abbreviation and name in extenso

- The aim of the paper, according to the title and to the abstract introduction, should be to assess the use of a new coil in emborrhoid procedure; this is not reported here.

M&M:

- the use of the novel coil should be an inclusion criteria

- Page 3, Lines 109-113: already reported in the introduction section, please delete

- Page 3, Line 13: “compassionate” why the authors use this term here? emborrhoid in this scenario, according to data reported, is not compassionate but aim to treat the patient definitely

- Please report if any patient suffered from hemorroids related to portal hypertension

- Page 3 Line 117: please spell the abbreviation FBS; other examples in the rest of the text; please check

- Outcomes measures: if the paper aim is “The primary objective of this study was to prospectively assess the safety and feasi-134 bility of treating grade I, II, and III hemorrhoids by superior rectal artery (SRA) emboliza-135 tion in a cohort of twenty-one patients.”, this manuscript does not present any novelty because this topic has been largely reported and discussed in literature; if instead the authors aim to report on a novel coil in the emborrhoid procedure, this should be assessed here. Please clarify here and in the rest of the manuscript

- Page 8 Line 221-222: sentence unclear; what the authors mean?

Discussion:

- Please focus also on clinical success

- Please report the advantage of the new coil and compare with other coils and embolics used for emborrhoids and reported in literature

- P10 Line 313-314: this sentence is unclear, please retype

- P10 Line 325: the correct name of the mentioned colleague is Iezzi and not “Lezzi”

- Conclusions, in this form, do not mention about the novelty of the paper, which is the use of the new coil; retype

Author Response

Point 1 The coil model should be detailed in the abstract.

Response 1: We agree with the recommendation and made the change. 

Point 2-In the M&M section of the abstract the authors report about 16 embolization procedures in 21 patients; this sounds strange and do not correspond in the numerical data of the rest of the paper. Please check.

Response 2: We recognize the error and indicate the number of 23 treated patients.

Point 3- In the results section of the abstract, after “significant” p-value should be reported

Response 3: We agree with the recommendation and make the change. We include the "p-values"

Point 4- In the conclusions section of the abstract, CDHD should be spelled being the first time it appears into the paper; furthermore the conclusions do not match the aim of the paper. Emborrhoids is a well known procedure; the novelty of the paper relies on the use of a novel coil. The authors should focus on the role of this new coil and not only on the outcome of the emborrhoid procedure.

Response 4: We agree with the recommendation. We highlight the characteristics and benefits of the new coil.

KEYPOINTS

Point 5- The second keypoint can not be derived from the study data; it seems to be an authors opinion rather than a study finding.

Response 5: The Catheter-directed hemorrhoidal dearterialization (CDHD) is a simple procedure, finding demonstrated as a technical result of the procedure.

Point 6 - Third keypoint: a coil is not a “closure device”, please correct with “embolic; furthermore this sentence should be limited to the emborrhoid procedure because this coil has been tested only in this scenario according to the data of the manuscript.

Response 6: We made the change by embolic material.

MANUSCRIPT

Point 7 - Many typewriting and grammar errors. English language should be improved; please refer to English mother-tongue speaker.

Response 7: The paper has undergone English language editing by MDPI , with its respective translation certificate.

Introduction:

Point 8 - Each time the authors mention a colleague, please use always the same style; for example surname abbreviation and name in extensor.

Response 8: We made the change.

Point 9- The aim of the paper, according to the title and to the abstract introduction, should be to assess the use of a new coil in emborrhoid procedure; this is not reported here.

Response 9: We agree with the recommendation. We highlight the characteristics and benefits of the new coil in the introduction.

M&M:

Point 10 - the use of the novel coil should be an inclusion criteria.

Response 10: The change was made.

Point 11 - Page 3, Lines 109-113: already reported in the introduction section, please delete.

Response 11: we delete the suggested

Point 12 - Page 3, Line 13: “compassionate” why the authors use this term here? emborrhoid in this scenario, according to data reported, is not compassionate but aim to treat the patient definitely.

Response 12: Most authors describe the procedure as "compassionate." because currently there is no significant evidence as a treatment of choice.

Point 13- Please report if any patient suffered from hemorroids related to portal hypertension.

Response 13: None of our patients had a history of cirrhosis or portal hypertension.

Point 14- Page 3 Line 117: please spell the abbreviation FBS; other examples in the rest of the text; please check.

Response 14: The corresponding revisions and corrections were made.

Point 15- Outcomes measures: if the paper aim is “The primary objective of this study was to prospectively assess the safety and feasi-134 bility of treating grade I, II, and III hemorrhoids by superior rectal artery (SRA) emboliza-135 tion in a cohort of twenty-one patients.”, this manuscript does not present any novelty because this topic has been largely reported and discussed in literature; if instead the authors aim to report on a novel coil in the emborrhoid procedure, this should be assessed here. Please clarify here and in the rest of the manuscript.

Response 15: The pertinent corrections were made, describing the results with the new novel coil.

Point 16- Page 8 Line 221-222: sentence unclear; what the authors mean?

Response 16: It describes where the survey used (Rørvik HD ) and the methodology for the evaluation of the perception of the state of health and the quality of life of the patients come from.

Discussion:

Point 17 - Please focus also on clinical success.

Response 17: The change was made.

Point 18 - Please report the advantage of the new coil and compare with other coils and embolics used for emborrhoids and reported in literature.

Response 18: The change was made.

Point 19 - P10 Line 313-314: this sentence is unclear, please retype.

Response 19: clarification is made.

Point 20 - P10 Line 325: the correct name of the mentioned colleague is Iezzi and not “Lezzi”

Response 20: The change was made.

Point 21 - Conclusions, in this form, do not mention about the novelty of the paper, which is the use of the new coil; retype.

Response 21: The pertinent corrections were made, describing the results with the new novel coil.

Round 2

Reviewer 1 Report

Points 5 and 8 have not been adequately addressed. 

The ideal comparison is with sclerotherapy. RBL as demonstrated in the suggested paper can be useful for prolapsing HD

Author Response

Response to Reviewer 1 Comments

Point 5: No guidelines or indications were used: Consensus statement of the Italian society of colorectal surgery (SICCR): management and treatment of hemorrhoidal disease. Tech Coloproctol. 2020 Feb;24(2):145-164. doi: 10.1007/s10151-020-02149-1.

Response 5: We added the Guideline proposed by reviewer 1 and added the following. “Guidelines on management and treatment of hemorrhoidal disease of the Italian Society were used. "Embolization was used in patients with disabling and refractory hemorrhoidal symptoms and without irreducible prolapse (Level of evidence: 2; Grade of recommendation: C).”

The corresponding bibliographic citation is added.

Gallo G, Martellucci J, Sturiale A, Clerico G, Milito G, Marino F, Cocorullo G, Giordano P, Mistrangelo M, Trompetto M. Consensus statement of the Italian society of colorectal surgery (SICCR): management and treatment of hemorrhoidal disease. Tech Coloproctol. 2020 ;24(2):145-164.

Point 8: Please compare your results with other similar procedures such as RBL (A Stepwise Proposal for Low-Grade Hemorrhoidal Disease: Injection Sclerotherapy as a First-Line Treatment and Rubber Band Ligation for Persistent Relapses. Front Surg. 2022 Jan 10;8:782800. doi: 10.3389/fsurg.2021.782800) or Sclerotherapy (A multicentre, open-label, single-arm phase II trial of the efficacy and safety of sclerotherapy using 3% polidocanol foam to treat second-degree haemorrhoids (SCLEROFOAM). Tech Coloproctol. 2022 Mar 25:1–10. doi: 10.1007/s10151-022-02609-w) and many others.

Response 8: We have added this paragrapf in the dissussion: Embolization treatment offers advantages over others such as Rubber band ligation (RBL) and injection sclerotherapy (IS), which can present post-procedure complications between 1-50% of RBL cases. Pain has been observed in 8–80%, and severe pain in 4–20%. (Tutino R 2022) In post-embolization treatment there is usually no post-procedure pain.

We Add this reference: Tutino, R., Massani, M., Jospin Kamdem Mambou, L., Venturelli, P., Della Valle, I., Melfa, G., Micheli, M., Russo, G., Scerrino, G., Bonventre, S., Cocorullo, G. . A Stepwise Proposal for Low-Grade Hemorrhoidal Disease: Injection Sclerotherapy as a First-Line Treatment and Rubber Band Ligation for Persistent Relapses. Frontiers in surgery 2022; , 8,782800

Reviewer 2 Report

the second version of the article shows many critical aspects:

in the abstract (line 26-27) the authors reported 26 procedures in 21 pts, but in the results they reported only 3 pts underwent second embolizations, so in total are 24! 

Goligher is a classification not a scale, and in table 1 and  line 71 it is written incorrectly.

line 51-52 CDHD is not a pathophysiological basis but a procedure.

line 76, bleeding is a symptom of hemorrhoidal disease  not a indication of hemorroidal disease.

Milligan Morgan are two surnames of two surgeons,  so it is correct Milligan Morgan not Milligan M.

line 118.  Emborrhoid is not a compassionate treatment.  Compassionate treatment means making a new, unapproved therapy available to treat a seriusly ill patient when no other treatments are available.

tab 2 it is not clear Cumulative % 68.7 and 80.9

tab 4 or Outcomes or Results,  no Outcome results...

it is important to always use the same abbreviations (SRA or RAS)

line 244-246 unclear

In results the Rorvik score is not reported , but in Materials and Metods  it is included.

line 294-295  Milligan Morgan and Fergouson described their hemorrhoidectomies. According to the litterature these techniques are considered the gold standard...  

Author Response

Point 1 In the abstract (line 26-27) the authors reported 26 procedures in 21 pts, but in the results they reported only 3 pts underwent second embolizations, so in total are 24! 

Response 1: There are effectively 24 procedures (21+3) in 21 patients. We correct it. Thank you.

Point 2- Goligher is a classification not a scale, and in table 1 and  line 71 it is written incorrectly

Response 2: Thank you. We correct it.

Point 3. Line 51-52 CDHD is not a pathophysiological basis but a procedure.

Response 3: We correct it and we change “Catheter-directed hemorrhoidal dearterialization (CDHD) is the procedure of embolization..”

Point 4- line 76, bleeding is a symptom of hemorrhoidal disease  not a indication of hemorroidal disease.

Response 4: We correct it and we change indication by symptom.

Point 5- Milligan Morgan are two surnames of two surgeons,  so it is correct Milligan Morgan not Milligan M.

Response 5: Thank you. We correct it Mulligan Morgan

Point 6 - line 118.  Emborrhoid is not a compassionate treatment.  Compassionate treatment means making a new, unapproved therapy available to treat a seriusly ill patient when no other treatments are available.

Response 6: The pioneer of this technique Dr Vidal et al have called it compassionate treatment and it is true that it is wrong as there are many better and worse treatments for hemorrhoidal disease. We remove the term and we change by alternative.

Point 7 - Tab 2 it is not clear Cumulative % 68.7 and 80.9

Response 7: We remove the cumulative data

Point 8 - tab 4 or Outcomes or Results,  no Outcome results..

Response 8: Thank you We correct it

Point 9- It is important to always use the same abbreviations (SRA or RAS)

Response 9: OK We are agree we Correct it

Point 10 - Line 244-246 unclear

Response 10: We have rewritten the paragraph.

Follow-up was performed with clinical assessment of symptoms and anoscopy at 3, 6 and 12 months.

Point 11. In results the Rorvik score is not reported, but in Materials and Metods  it is included.

Response 11: We included table with results

Point 12. Line 294-295  Milligan Morgan and Fergouson described their hemorrhoidectomies. According to the litterature these techniques are considered the gold standard... 

Response 12: Thank you We correct it

Reviewer 3 Report

Authors properly responded to my suggestions.

Author Response

Point 1. Authors properly responded to my suggestions.

Response 1: Thanks.

Round 3

Reviewer 1 Report

I'm satisfied with the suggested changes